# Design of Flow Velocity and Direction Monitoring Sensor Based on Fiber Bragg Grating

**DOI:** 10.3390/s21144925

**Published:** 2021-07-20

**Authors:** Hao Zhang, Zhixin Zhong, Junmiao Duan, Guangxun Liu, Junhai Hu

**Affiliations:** 1State Key Laboratory of Mechanical Behavior and System Safety of Traffic Engineering Structures, Shijiazhuang Tiedao University, Shijiazhuang 050043, China; zhanghao@stdu.edu.cn; 2Structural Health Monitor and Control Institute, Shijiazhuang Tiedao University, Shijiazhuang 050043, China; 3School of Civil Engineering, Shijiazhuang Tiedao University, Shijiazhuang 050043, China; zhongzhixinsz@163.com (Z.Z.); liuguangxun0518@163.com (G.L.); 4School of Traffic & Transportation, Shijiazhuang Tiedao University, Shijiazhuang 050043, China; 5State Key Laboratory of Traction Power, Southwest Jiaotong University, Chengdu 610031, China; hujunhai2215@163.com

**Keywords:** fiber Bragg grating, flow velocity and direction, simulation analysis, calibration experiment

## Abstract

The real-time monitoring of the flow environment parameters, such as flow velocity and direction, helps to accurately analyze the effect of water scour and provide technical support for the maintenance of pier and abutment foundations in water. Based on the principle of the Fiber Brag Grating sensor, a sensor for monitoring the flow velocity and direction in real-time is designed in this paper. Meanwhile, the theoretical calculation formulas of flow velocity and direction are derived. The structural performance of the sensor is simulated and analyzed by finite element analysis. The performance requirements of different parts of the sensor are clarified. After a sample of the sensor is manufactured, calibration experiments are conducted to verify the function and test the accuracy of the sensor, and the experimental error is analyzed. The experimental results indicate that the sensor designed in this paper achieves a high accuracy for the flow with a flow velocity of 0.05–5 m/s and the flow velocity monitoring error is kept within 7%, while the flow direction monitoring error is kept within 2°. The sensor can meet the actual monitoring requirements of the structures in water and provide reliable data sources for water scour analysis.

## 1. Introduction

In the long-span bridge construction and operation process, the construction platform and pier foundation are affected by water scouring and other factors [1,2,3,4]. This can cause dynamic softening of pier foundation [5], material erosion and aging, and attenuation of structural components and overall resistance, thus affecting the safety and durability of the structure [6,7]. Real-time monitoring of flow velocity and direction is essential to effectively analyze and control the scour of water flow on the construction platform and pier, reasonably maintaining the pier foundation and abutment in water and ensuring safety during construction and operation [8,9]. 

At present, the monitoring instrument of ocean flow environment parameters, such as flow velocity and flow direction, mainly imports the parameters to the Acoustic Doppler Current Profiler (ADCP). Based on the well-known physical principle of Doppler frequency shift, the ADCP instrument obtains the velocity of the corresponding point by measuring the Doppler frequency shift. However, this instrument has the drawbacks of high cost and complex circuit integration, and it is easily affected by the external environment. Sensors that exploit electromagnetic waves and the acoustic Doppler principle to measure the flow velocity and direction have become popular [10]. The measurement accuracy of these sensors is relatively high, but the manufacturing cost is high. Meanwhile, the sensors are vulnerable to electromagnetic environment interference, and the measurement result is not ideal, which limits the application scenarios of these sensors [11].

In recent years, Fiber Bragg Grating (FBG) has been gradually used as a measuring element of various sensors because of its advantages such as being not easily interfered with by the external electromagnetic environment [12,13], high measurement accuracy [14,15], small volume, good wavelength selectivity [16], and not being affected by nonlinear effects. However, considering the fiber Bragg grating sensing technology, most scholars still study the flow velocity monitoring of directional fluid [17]. For specific bridge piers, there are few reports on the application of this technology to the nondirectional monitoring of flow velocity and direction [18,19,20,21,22]. At present, FBG sensors have been widely accepted in China, and the project client has designated the use of FBG sensors. Therefore, an FBG sensor for flow velocity and flow direction monitoring is proposed in this paper to provide technical support for bridge construction and operation.

### Sensor Structure

The proposed sensor consists of two parts: the upper part and the lower part. The upper part is mainly composed of sensing elements such as frame base, cam, spring, and metal wire. The lower part is mainly composed of a cardan plate and a steering shaft. As shown in Figure 1, the components of the sensor are listed as follows: (1) venturi, (2) sliding connection, (3) eccentric wheel, (4) washer, (5) closed push rod, (6) displacement conduction shaft, (7) displacement top shaft, (8) strain grating, (9) wire, (10) nut, (11) steering shaft, (12) simple beam, (13) support frame, (14) frame base, (15) rolling bearing, (16) coding plate of shaft sleeve, (17) water turbine, (18) lever, (19) universal plate, (20) coding plate of shaft sleeve and (21) optical fiber Grating.

## 2. Principle of Sensor Measurement

### 2.1. Flow Direction Measurement Principle

The sensor performs flow direction measurements in the following three steps.

#### 2.1.1. Universal Plate for Adaptive Water Flow

To solve the problem of non-orientation flow direction, the sensor needs to adapt to the change of flow direction. The adaptive function of the sensor is supported by the universal board. Specifically, one side of the universal board is fixed on the rotating shaft so that the board can rotate with the rotating shaft, and the rotating bearing is fixed on the frame. When the sensor works, the rotating shaft is vertical and the cardan plate rotates in the horizontal plane with the change of the flow direction. In this way, the cardan plate drives the rotating shaft to rotate together. When the flow direction is parallel to the plane of the cardan plate and the flow is from the near shaft end to the far shaft end, the cardan plate will be in the balance state and will not rotate; otherwise, it will continue to rotate to the balance state. The schematic diagram of the universal plate is shown in Figure 2. The universal plate is in the same plane as the water flow to enable the sensor to adapt to the change of flow direction.

#### 2.1.2. Flow Direction Conversion Principle

(1) The flow direction is converted to the rotation of the shaft.

The cardan plate is fixedly connected with the rotating shaft. In the process of rotation, the cardan plate drives the rotating shaft to rotate together. Thus, the problem of fluid flow direction can be transformed into the problem of turning the shaft.

(2) The rotation direction of the rotating shaft is converted into the displacement of the closed push rod.

As shown in Figure 3, eccentric wheel 1 is installed on the rotating shaft. The distance from the edge of the eccentric wheel 1 to the center of the rotating shaft corresponds to the turn of the rotating shaft. When the instantaneous eccentric wheel 1 rotates clockwise with the rotating shaft, the closed push rod 1 moves horizontally to the right; when the rotating shaft rotates normally, the rotation direction is uncertain, but the closed push rod 1 only moves left and right at the position shown in Figure 3. When rotating, the eccentric 1 pushes the closed push rod 1 at different distances away from the center of the shaft, and different distances correspond to different directions.

#### 2.1.3. Realizing 360° Flow Direction Monitoring

(1) Setting of eccentric 2 and closing push rod 2.

The sensor can only distinguish the direction of 180° based on the single eccentric wheel 1 because the axis passing through the center of the shaft is symmetric. In the monitoring process, the sensor needs to accurately identify the direction of water flow and distinguish the flow direction of 360° on a certain plane. Therefore, an eccentric wheel 2 and a closed push rod 2 are added to the sensor, as shown in Figure 4. The assembly position of the eccentric wheel 2 deviates 90° from that of the eccentric wheel 1 (the eccentric wheels 1 and 2 have the same size and shape). In this way, when the eccentric wheel 1 causes the same displacement of the closed push rod 1 in the symmetrical position, another displacement of the closed push rod 2 caused by the eccentric 2 can help distinguish the symmetry of the eccentric 1. 

(2) Flow direction conversion principle of cam push rod mechanism

Figure 5 and Figure 6 illustrate the diagrams of the two groups of eccentric wheels and closed push rods in the sensor. $O_{1}$ and $O_{2}$ are the rotation centers, while $O_{1}^{\prime }$ and $O_{2}^{\prime }$ are the centers of the two eccentric wheels with the same size and shape. The assembly is conducted by passing the rotating shaft through the eccentric wheel.

The shaft and the eccentric wheel are fixed by the key in the keyway, and the upper part and lower part of the eccentric wheel are separated by washers. During the installation, the assembly position of the eccentric wheel 1 and the eccentric wheel 2 deviates 90°. As for the eccentric wheel 1 at position 1 and position 2, it can be seen from the symmetry that the displacement of the closed push rod 1 is the same as that of position 2, and the corresponding direction of the eccentric wheel 1 at position 1 and position 2 cannot be distinguished. By contrast, the displacement of the closed push rod 2 caused by the eccentric wheel 2 at position 1 is smaller than that caused by the eccentric wheel 2 at position 2. Therefore, the displacement of the closed push rod 2 can be exploited to distinguish the flow direction of 360°. 

(3) Transformation of closed push rod displacement into FBG strain [23].

The spring is connected with the metal wire at both ends of the closed push rod and the frame. As shown in Figure 7, the FBG is pasted on the metal wire to sense the tension and compression of the beam. Two metal wires and three FBGs are, respectively, set at the left and right of the metal wire. By measuring the wavelength change of the FBGs at both ends, the water flow within the direction of 360° can be monitored.

### 2.2. Principle of Velocity Measurement

The sensor performs velocity measurements in the following four steps.

#### 2.2.1. Cardan Plate Flow Direction Adaptation

According to the adaptive water flow of the Cardan plate introduced in Section 2.1, when the water flow direction is parallel to the plane of the Cardan plate and flows from the near shaft end to the far shaft end, the cardan plate will finally be in the balance state and will not rotate; otherwise, it will continue to rotate to the balance state. The schematic diagram of the universal plate for adaptive water flow is shown in Figure 8.

#### 2.2.2. Velocity Sensitization and Amplification of Venturi

After the water flow adapts itself, the water flows in from the venturi inlet and out from the contraction outlet. According to the principle of equal flow, the accurate measurement of low flow velocity can be achieved by enlarging the flow velocity and increasing the measurement sensitivity.

#### 2.2.3. Conduction Velocity of Water Turbine under Stress

The water flows out from the contraction port and contacts the water turbine. In this way, the water turbine is driven to rotate, and the lever transmits the rotation change of the water turbine to the displacement conduction shaft, driving the displacement top shaft to move up and down and filter out the rotation direction of the displacement conduction shaft. As shown in Figure 9, the change of the water flow velocity causes the frequency of the periodic reciprocating motion of the displacement top shaft to change.

#### 2.2.4. Transmit Velocity into Grating Frequency

The displacement conducting shaft passes through the hollow steering shaft, and it is connected with the upper rotating bearing and the displacement top shaft. The upper end of the displacement top shaft is also connected with the simply supported beam. The FBG is pasted on the middle span of the upper end of the simply supported beam with a special flexible material (e.g., epoxy resin). To eliminate the influence of temperature on measurement, a temperature compensation grating is connected in series. As shown in Figure 10, the reciprocating driving force of the displacement top axis causes the simply supported beam to vibrate, which results in the change of the wavelength frequency of the grating. Besides, the upper structure is installed into the circular protective cylinder as a whole. Based on this, the water infiltration that affects the test in the measurement process can be avoided. 

#### 2.2.5. Grating Frequency Identification Method

In the experiment, the FBG demodulator is exploited to set the sampling frequency and collect the wavelength data of the FBG and the temperature compensation grating. The initial central wavelength is subtracted from the collected wavelength, and the obtained wavelength variations are denoted as $\Delta \lambda_{1}$ and $\Delta \lambda_{2}$. Then, a difference between $\Delta \lambda_{1}$ and $\Delta \lambda_{2}$ that is greater than 0.1 nm is the frequency value corresponding to the flow rate.

### 2.3. Calculation Formula, Measuring Range and Accuracy of the Sensor

#### 2.3.1. Theoretical calculation and accuracy of flow direction

(1) Theoretical calculation of flow direction.

The theoretical calculation of the wavelength variation and strain transformation of FBG is presented in Formula (1):(1)$$\Delta \lambda_{B}=\lambda_{B}(1-P_{\varepsilon })\varepsilon ,$$
where $\Delta \lambda_{B}$ is the wavelength change of the FBG (mm); $\lambda_{B}$ is the original center wavelength of the FBG (nm); $P_{\varepsilon }$ is the effective elastic optical coefficient (0.22); and ε is the strain of the tested object.

Based on this, the following two formulas can be obtained:(2)$$\varepsilon_{1}=\frac{\Delta \lambda_{B1}}{(1-P_{\varepsilon })\lambda_{B1}},\;\varepsilon_{2}=\frac{\Delta \lambda_{B2}}{(1-P_{\varepsilon })\lambda_{B2}},$$
where $\varepsilon_{1}$ and $\varepsilon_{2}$ are the strain measured by the FBG attached to the wires 1 and 2, respectively; $\Delta \lambda_{B1}$ and $\Delta \lambda_{B2}$ are the wavelength changes of the FBG; $\lambda_{B1}$, and $\lambda_{B2}$ are the original center wavelengths of the FBG (nm).

The position changes of the eccentric wheels 1 and 2 are shown in Figure 11.

The displacement of the push rod caused by the eccentric wheels 1 and 2 can be obtained from the geometric relationship:
(3)$$X_{1}=e(1-\cos\alpha ),\;X_{2}=e\sin\alpha ,$$
where is the eccentricity of eccentric (mm);  is the angle of eccentric rotation (°); and are the horizontal displacement produced by the two push rods.

The horizontal displacement of the two push rods is theoretically equal to the change of the spring and the wire. Thus, the external force on the two wires can be calculated through Formula (4). By applying Formulas (3) and (4) to Formula (2), Formulas (5) and (6) can be obtained.
(4)$$F_{1}=\frac{\Delta \lambda_{B1}EA}{\lambda_{B1}(1-P_{\varepsilon })},F_{2}=\frac{\Delta \lambda_{B2}EA}{\lambda_{B2}(1-P_{\varepsilon })}$$
(5)$$X_{1}=\frac{\Delta \lambda_{B1}l}{\lambda_{B1}(1-P_{\varepsilon })}-\frac{\Delta \lambda_{B1}EA}{\lambda_{B1}(1-P_{\varepsilon })K}$$
(6)$$X_{2}=\frac{\Delta \lambda_{B2}l}{\lambda_{B2}(1-P_{\varepsilon })}-\frac{\Delta \lambda_{B2}EA}{\lambda_{B2}(1-P_{\varepsilon })K},$$
where $F_{1}$ and $F_{2}$ are the spring force on the two wires; K is the spring stiffness (N/mm^2^); l is the wire length (mm); and E is the modulus of elasticity of the wire (Kpa). A is the stress section area of the wire (mm^2^).

Based on Formulas (2) to (6), it can be obtained that:(7)$$\cos\alpha =1-(\frac{\Delta \lambda_{B1}l}{\lambda_{B1}(1-P_{\varepsilon })e}+\frac{\Delta \lambda_{B1}EA}{\lambda_{B1}(1-P_{\varepsilon })Ke})$$
(8)$$\sin\alpha =(\frac{\Delta \lambda_{B2}l}{\lambda_{B2}(1-P_{\varepsilon })e}+\frac{\Delta \lambda_{B2}EA}{\lambda_{B2}(1-P_{\varepsilon })Ke})$$
(9)$$\alpha =\arctan\frac{\lambda_{B1}(EA+Kl)\Delta \lambda_{B2}}{\lambda_{B2}[Ke(1-P_{\varepsilon })\lambda_{B1}-(EA+Kl)\Delta \lambda_{B1}]}$$

Finally, the flow direction α is determined by values of sinα and cosα.

(2) Theoretical accuracy of the flow direction.

The theoretical strain of the FBG is $\varepsilon =\frac{F}{EA}$. Meanwhile, the sum of the spring deformation and wire deformation is equal to the displacement of the push rod. Thus, $X=\frac{\Delta \lambda_{B}l}{\lambda_{B}(1-P_{\varepsilon })}-\frac{\Delta \lambda_{B}EA}{\lambda_{B}(1-P_{\varepsilon })K}$. According to Formulas (4) and (5), the accuracy of the flow direction measurement is as follows:(10)$$S_{1}=\frac{\Delta \lambda_{B1}}{\sin\alpha }=\frac{Ke(1-P_{\varepsilon })\lambda_{B1}}{EA+Kl}$$
(11)$$S_{2}=\frac{\Delta \lambda_{B1}}{1-\cos\alpha }=\frac{Ke(1-P_{\varepsilon })\lambda_{B2}}{EA+Kl},$$
where *S* is the sensor accuracy.

It can be seen from the above formulas that there are two expressions for the theoretical accuracy of the sensor, but they all depend on the parameters of the material. Therefore, the maximum value of the two expressions should be taken as the final accuracy.

#### 2.3.2. Velocity Calculation, Range, and Accuracy

(1) Velocity calculation.

Considering the fluid flowing into the venturi at a velocity of $V_{1}$, according to the hydrodynamics and the principle of equal flow rate, we have: (12)$$V_{1}\times S_{1}=V_{2}\times S_{2}$$

It can be seen that the velocity of the fluid magnified by the venturi is $V_{2}$. As for the fluid flowing through the blades at this speed, the speed of the contact point between the water flow and the turbine is also $V_{2}$. According to the principle of the equal angular velocity of coaxial rotation, it can be known that the linear velocity of the pinion rotation is $V_{3}$.

It is assumed that the number of pinion teeth is N. Since one gear corresponds to a reciprocating movement of the displacement top axis, the movement of the displacement top axis causes the simply supported beam to vibrate. The FBG demodulator can obtain the frequency f of the wavelength change of the FBG caused by the vibration of the simply supported beam. According to f=1/T, the time interval T of a reciprocating vibration of the displacement top axis can be calculated. Thus, we have:(13)$$T\times N\times V_{3}=2\pi R_{2}.$$

The relationship between the primary vibration of the displacement top shaft and the arc length of pinion rotation can be seen in Equation (13). Finally, the calculation formula of flow velocity $V_{1}$ is derived as follows:(14)$$V_{1}=\frac{2\pi R_{1}S_{2}}{NS_{1}}\times f.$$

In Formulas (12)–(14), $V_{1}$ is the velocity of the fluid flowing into the venturi (m/s); $R_{1}$ is the radius of the water wheel (mm); $V_{2}$ is the velocity of the fluid flowing out of the venturi (m/s); $R_{2}$ is the radius of the pinion (mm); $V_{3}$ is the cog speed (m/s); T is the vibration period of the cam push rod (s); $S_{1}$ is the initial cross-sectional area of the fluid flowing into the venturi (mm^2^); N is the number of the pinion gears (individual); $S_{2}$ is the cross-sectional area of the fluid flowing out of the venturi (mm^2^); and f is the vibration frequency of the displacement top shaft (Hz).

Based on the established mathematical model, the corresponding flow velocity can be calculated by monitoring the wavelength change frequency of the grating, thus realizing the dynamic measurement of the flow velocity.

(2) Sensor range and accuracy analysis.

According to the theory of hydrodynamics, when the fluid flows through the turbine blade at the speed V, the blade will rotate under the friction resistance $F_{f}$. The calculation of $F_{f}$ is shown in Formula (15).
(15)$$F_{f}=C_{f}\times p\times \frac{V^{2}}{2}\times B\times L.$$

Applying Formula (12) to Formula (15), $F_{f}$ can finally be obtained, as shown in Equation (16).
(16)$$F_{f}=C_{f}\times p\times \frac{V_{1}{}^{2}S_{1}{}^{2}}{2S_{2}^{2}}\times B\times L$$

In Formulas (15) and (16), $F_{f}$ is the friction resistance of the turbine blade (N), and p is the density of the flow (1000 Kg/m^3^); $C_{f}$ is the dimensionless resistance coefficient of flow viscosity, and an empirical value of this coefficient is 1.3 × 10^−6^; *B* is the width of the turbine blade (mm); and *L* is the length of a single blade of hydraulic turbine along the direction of fluid movement (mm).

The minimum speed of water turbine rotation, that is, the minimum range of flow velocity measured by the sensor, can be obtained from Formula (16). However, due to the different processing technology of the sensor, the formula can only be used to calculate the theoretical minimum flow rate, and the actual minimum flow rate should be obtained through the actual calibration experiment.

According to Formula (14), the measurement accuracy of the sensor for the flow rate is:(17)$$S=\frac{\Delta f}{\Delta V}=\frac{NS_{1}}{2\pi R_{1}S_{2}}.$$

It can be seen from Equation (17) that the measurement accuracy of the sensor depends on the ratio of the cross-sectional area of the venturi inlet end to the outlet end and the radius of the water wheel as well as the number of teeth of the pinion. The smaller the ratio, the higher the measurement accuracy of the sensor. However, the actual accuracy should be based on the sensor performance calibration experiment. According to the above theoretical analysis and theactual monitoring requirements of the sensor, the dimensions of the venturi and water wheel are shown in Figure 12 and Figure 13, respectively.

## 3. Simulation Analysis of Key Parts of the Sensor

### 3.1. Finite Element Stress Analysis of the Sensor Hydraulic Structure

Finite element stress analysis of water turbine is conducted to select the water turbine materials reasonably [24]. Specifically, the center of the water turbine is fixed. Then, according to Formula (17), a fluid load interface of 10 N is constructed at the bottom gear. Though the fluid load interface has a calculated value of 8.65 N, a value of 10 N is used to simplify the model. After the gear on the water wheel starts to rotate, the stress and strain of the wheel are extracted, which are illustrated in Figure 14 and Figure 15, respectively.

According to the stress and strain diagrams of the water wheel, the connection between the gear and the water wheel has obvious stress and large strain, and there is almost no stress on other parts of the water wheel. Considering the processing difficulty, the same material should be selected for the overall structure of the water turbine, especially the material with high strength and sensitive response. If the water wheel needs to be immersed in water for a long time, the material with excellent corrosion resistance should be selected.

### 3.2. Fluid–Solid Coupling Analysis of the Venturi in the Sensor

The water flowing through the venturi has an impact on the venturi. If the impact force of the water flow is too large, the necking section of the venturi will be deformed. Thus, it is necessary to analyze the internal stress and strain of the venturi. The venturi has certain friction to the flow and reduces the flow velocity. Therefore, it is also necessary to analyze the magnitude of the friction of the venturi and the reduction of flow velocity.

Firstly, a venturi model is established in gambit. Then, the fluid–solid coupling analysis is conducted in the finite element analysis software to obtain the change of the stress–strain and flow velocity of the venturi. Subsequently, the interaction between these two factors is found [25,26]. The results of stress and strain are illustrated in Figure 16 and Figure 17, and the change of flow velocity is shown in Figure 18.

It can be seen from Figure 16 and Figure 17 that the stress at the necking section of the venturi is the largest and the strain obviously changes. According to the design size and amplification principle of the venturi, the flow velocity of the fluid flowing through the venturi is increased by 13.3 times. It is observed from Figure 18 that the final flow velocity is 11.71 times greater than the initial one.

According to the above analysis, the stress of the flow at the necking section of the venturi changes due to the necking of the section, and the increase of the flow velocity causes a large strain change at the necking of the inner wall of the venturi. Meanwhile, the flow velocity is affected by the friction from the inner wall of the venturi, resulting in the reduction of the flow velocity by 1.59 times. As for selecting the material of venturi, the material with high strength, low friction resistance, and corrosion resistance should be selected. In the later improved design, the necked section of the venturi should be as smooth as possible with small friction resistance to avoid the sudden change of the section.

## 4. Experimental Analysis

The sensor is processed by combing 3D printing technology and precision machining. The real object is shown in Figure 19, and the calibration scheme shown in Figure 20 is used for the experiment.

In this experiment, the latest NSZ-FBZ-A06 dynamic fiber grating demodulator from China Nanzhi Sensor Technology Co., Ltd. was used for monitoring. The 8-channel measurement frequency was 5 KHz, and the single channel measurement frequency was 50 KHz. Five uniform FBGs were used as strain measurement and temperature compensation gratings. The parameters of these FBGs are listed in Table 1.

Note: In this paper, two temperature compensated FBG sensors were set up. One sensor was used to compensate for the effect of temperature on the wavelength change, i.e., compensation in the flow direction measurement. When the metal strip is pulled and pressed by the spring, the sensing force of the FBG pasted on the metal strip changes, and the temperature compensated grating pasted on other fixed positions without force compensates for the wavelength change caused by the sensing temperature of the fiber Bragg grating. The other sensor was used to investigate the effect of temperature on the wavelength frequency when measuring the flow rate. It was found that the temperature has little effect on the frequency, so the other temperature compensation grating has little effect.

### 4.1. Analysis of Flow Direction

#### 4.1.1. Relationship between the Test Wavelength and the Theoretical Flow Direction

After the test of the two experimental wavelengths was processed and analyzed, the relationship between the wavelength and flow direction is shown in Figure 21. In this experiment, a one dimensional parameter orthogonal wavelet was used to denoise the vibration signal and remove the variation of wavelength caused by temperature change. Meanwhile, the flow direction from the steering shaft end to the cardan plate is regarded as 0°, and the clockwise rotation as positive. This study of the cardan plate was in the direction of parallel and positive vertical water flow (0–90°). According to the positive and negative values of sinα and cosα, the flow direction of another 270° can be obtained, which will not be described here.

It can be seen from Figure 21 that the wavelength increment of the four fiber gratings increases gradually with the flow direction, which is consistent with the fact that the two metal wires are pulled to increase the wavelength during the measurement of the flow direction from 0° to 90°. For test 1, when the angle of the flow direction is small, the change of wavelength 1 is greater than that of wavelength 2; when the flow direction increases gradually, it turns out just the opposite. For test 2, when the angle of the flow direction is small, the change of wavelength 1 is larger than that of wavelength 2; when the flow direction increases, the initial change of wavelength 1 is smaller than that of wavelength 2, but the final change of wavelength 1 is larger than that of wavelength 2.

When the sensor is applied to monitor the change of water flow direction, the left wire in the sensor begins to be pulled, so the change of wavelength 1 is slightly greater than that of wavelength 2 at the beginning; when the flow direction increases gradually, the change of the two wavelengths tends to be the same. However, due to the pasting technology and the pretension strength of the FBGs, when the flow direction increases to 70–90°, the changes of the two wavelengths are different.

#### 4.1.2. Relationship between the Test Wavelength and the Theoretical Flow Direction

The error between the theoretical values and the experimental standard values was calculated, and the error percentage is shown in Figure 22. 

It can be seen from Figure 22 that the error of test 2 fluctuates more than that of experiment 1, but the error percentage of the two flow direction experiments is always kept within 2°, showing better experimental accuracy. The error also decreases gradually with the increase of the flow direction change. The test error of the sensor meets the requirements of the existing flow direction specification.

### 4.2. Experimental Analysis of Velocity

A fitting analysis was conducted on the flow velocity monitoring, and the result was compared with the theoretical frequency, which is shown in Figure 23. Referring to the monitoring range of common flow velocity sensors and the flow velocity monitoring requirements of the cross-sea bridges in world, the experiment reduced the noise of the vibration signal and then extracted the experimental data of 0.05–5 m/s following the method in Section 2.2.5.

It can be seen from Figure 23 that the frequency values of the two experiments are mainly distributed above the theoretical ones, and the fitting analysis of the experimental data obtains a linear line, that is, y=0.0006x. Thus, there is a linear relationship V=0.0006f between the monitoring data of the sensor and the actual measured flow rate. Since it can be seen from Formula (14) that the theoretical relationship between the flow rate and the frequency is V=0.0005895f, there are some errors between the experimental curve and the theoretical curve.

The analysis result of the error percentage between the experimental frequency and the theoretical frequency is shown in Figure 24. According to the error range of flow velocity, there is a certain error between the frequency calculated theoretically and the frequency monitored actually, but the error is small and is always kept within 7%, indicating that the sensor has a high-precision test in the flow velocity range of 0.05 m/s to 5 m/s.

### 4.3. Experimental Error Analysis

#### 4.3.1. Experiment Error of Flow Direction Calibration 

The experimental error of flow direction calibration mainly comes from the following aspects:The lubrication degree between the two eccentric wheels: two eccentric wheels and two closed push rods conduct the flow direction. Insufficient lubrication between the eccentric wheels may cause the flow direction transmission to be blocked, thus affecting the accuracy of the test.The strength of the metal wire: the monitoring grating used in this sensor is pasted on the metal wire. If the strength of the metal wire is not uniform enough, it will easily weaken the FBG’s ability to sense strain change and cause errors. In the displacement measurement of the push cam mechanism, the spring and wire are welded to avoid the delay phenomenon in the process of force transmission. However, the connection mode of the spring and wire may affect the accuracy of the measurement, and this problem will be further studied in the future.The pasting technology and pretension setting of FBGs: because the metal wire may be pulled or pressed, stress and strain are on FBG at the same time. A better pasting technology and enough pretension are needed to accurately reflect the strain change on the metal wire. Therefore, the pasting technology and pretension setting of FBGs also lead to experimental errors.

#### 4.3.2. Experimental Error of Velocity Calibration

The experimental error of velocity mainly comes from the following aspects:Sensitivity of the water wheel: the sensor senses force mainly by the rotation of the water wheels. However, the transfer efficiency of the water wheels could not reach 100% of the ideal state. The more sensitive the water wheels, the more accurate the sensor measurement data. The water wheel used in this experiment was processed by 3D printing technology, and the sensitivity was not high enough. In this case, it results in a low-precision test for a large flow velocity.Machining accuracy of the sensor’s overall structure: the processing technology and method used by the sensor seriously affect the test performance of the sensor. The more precisely the sensor is processed, the better the measurement performance is. So, the machining accuracy is one of the factors affecting the experimental error of the sensor.Sensitivity of the simply supported beam: the monitoring grating used by the sensor was pasted on the simply supported beam. If the sensitivity of the simply supported beam is not good enough, the intensity is not uniform enough, which will easily weaken the FBG’s ability to sense the change of vibration frequency and produce experimental errors. It will also be affected by the mechanical resonance of the simply supported beam. If the measurement is kept at a high speed, the performance and the measurement data of the sensor will be affected.

## 5. Conclusions and Prospect

Aiming at the demand for bridge piers and abutments, this paper designs a new type of sensor to monitor the flow direction and velocity. This paper also introduces the monitoring principle of the sensor in detail and deduces the theoretical calculation formulas of the flow direction. Simulation analysis, calibration experiments and error analysis of the designed sensor are conducted. The conclusions of this paper are drawn as follows: Based on the FBG sensing technology, the sensor realizes 360° flow direction self-adaptive monitoring and forward flow measurement. By enlarging the flow velocity, the sensor realizes an accurate measurement for a low flow velocity of 0.05–5 m/s, and the flow velocity monitoring error is kept within 7%, while the flow direction monitoring error is kept within 2°. Thus, the sensor can meet the requirements for monitoring the flow velocity and the direction of the water environment where cross-sea bridges and other structures are located.According to the stress analysis of the water wheel in the sensor by finite element analysis, the materials with high strength, sensitive stress and strong corrosion resistance should be selected to produce the water wheel; according to the results of fluid–solid coupling analysis of the venturi by finite element analysis, it is concluded that the inner wall of the venturi will have a friction effect on the flow, resulting in a reduction of the flow velocity of the venturi outlet by about 1.59 times compared with the theoretical value. Therefore, future sensor processing should improve the smoothness of the necking section in the venturi.According to the analysis of experimental error sources, the test accuracy is affected by the following factors: insufficient lubrication of the eccentric wheel and closed push rod, the machining accuracy of the sensor’s overall structure, the strength of the metal wire, and the pasting technology and pretension setting of the FBGs [15,16].

Therefore, at a later stage, we will simplify the sensor design and adopt different types of FBGs and pasting technology to improve the measurement accuracy of the sensor.

## Figures and Tables

**Figure 1 sensors-21-04925-f001:**
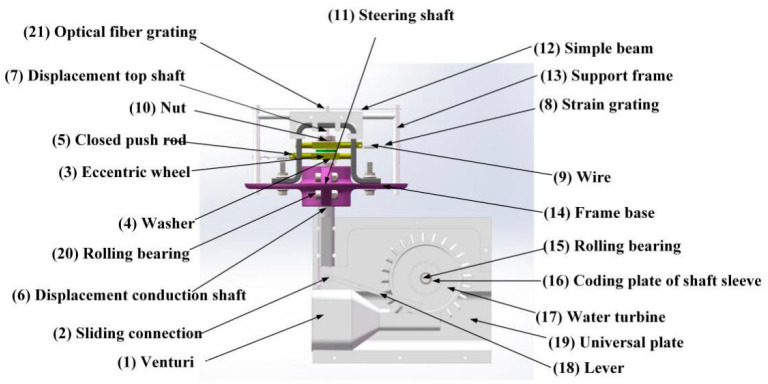
The structure of Flow Velocity and Direction Sensor.

**Figure 2 sensors-21-04925-f002:**
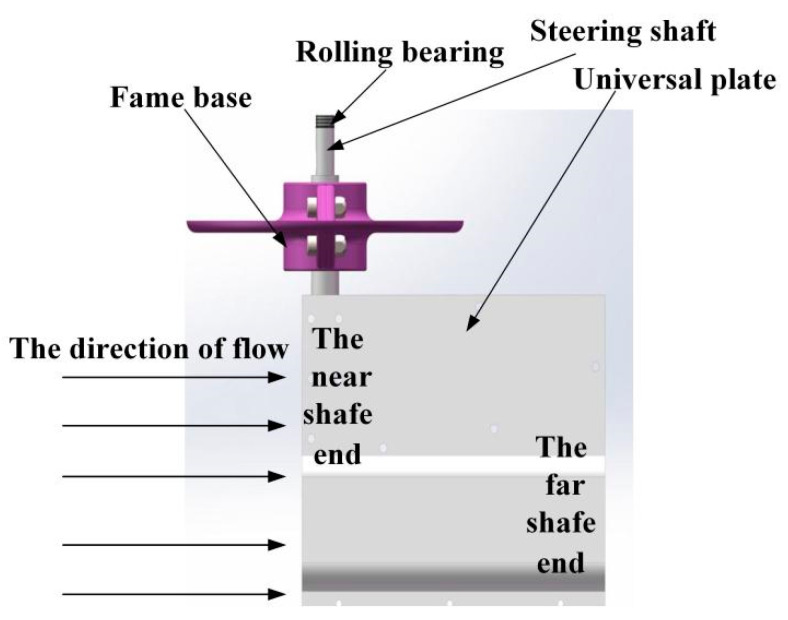
Universal plate adaptive flow.

**Figure 3 sensors-21-04925-f003:**
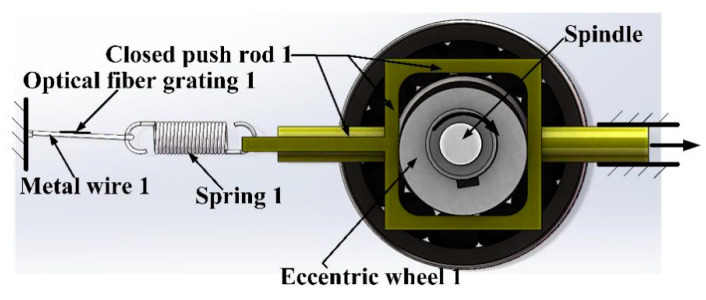
The displacement measurement of the push cam mechanism.

**Figure 4 sensors-21-04925-f004:**
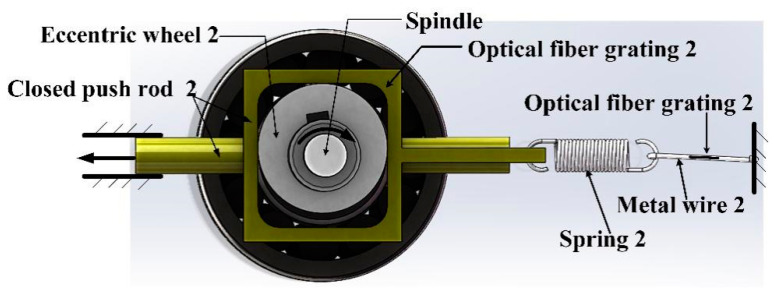
The displacement measurement of eccentric wheel two offset 90° and enclosed push rod two.

**Figure 5 sensors-21-04925-f005:**
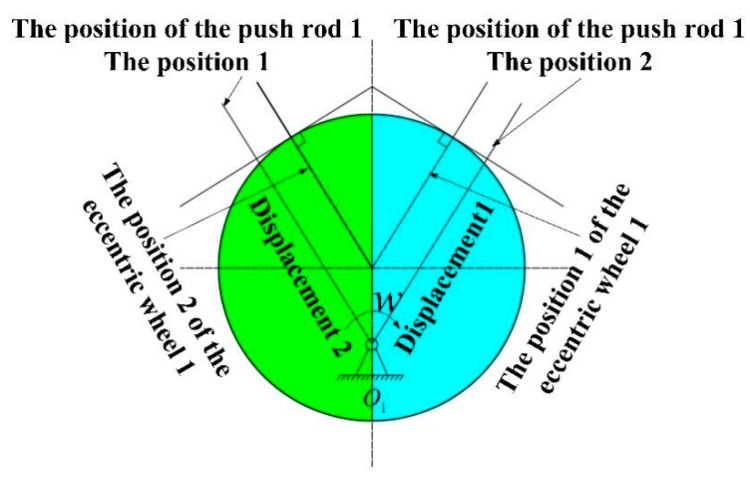
Push cam mechanism one.

**Figure 6 sensors-21-04925-f006:**
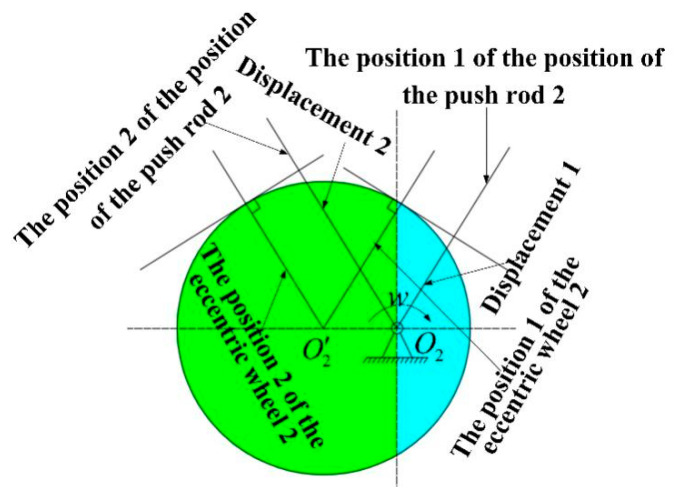
Push cam mechanism two.

**Figure 7 sensors-21-04925-f007:**
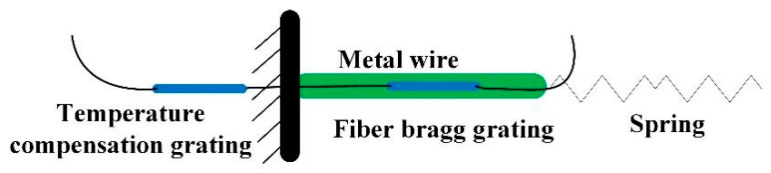
The fiber grating paste.

**Figure 8 sensors-21-04925-f008:**
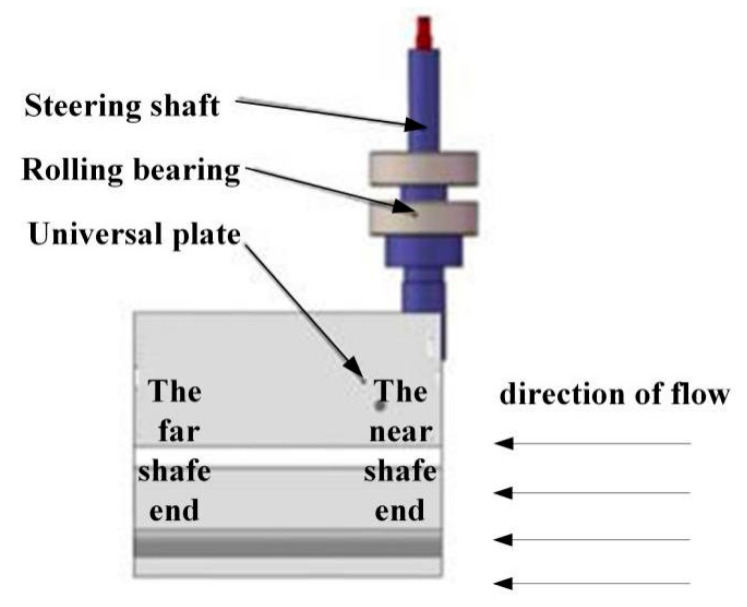
The universal plate of Sensor adaptive water flow.

**Figure 9 sensors-21-04925-f009:**
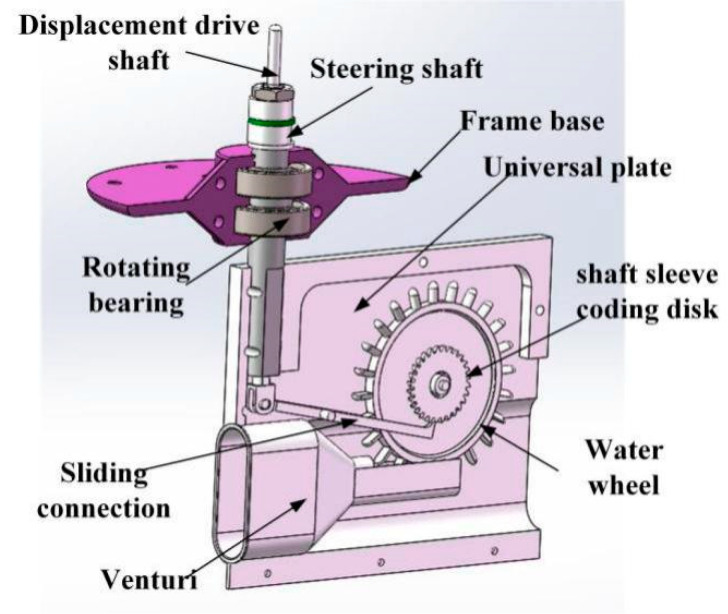
Principle of velocity measurement.

**Figure 10 sensors-21-04925-f010:**
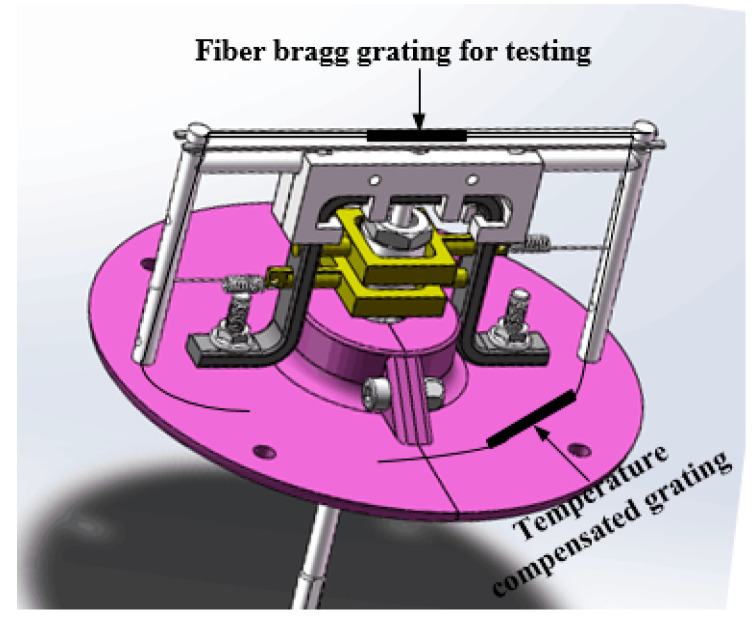
Fiber Bragg Grating paste position.

**Figure 11 sensors-21-04925-f011:**
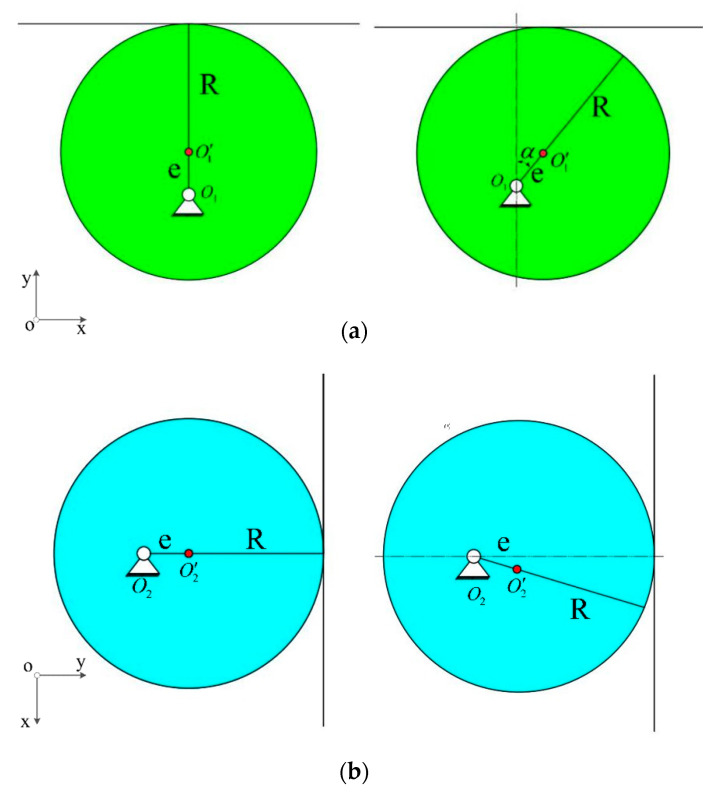
(**a**) The position of the eccentric wheel 1; (**b**) the position of the eccentric wheel 2.

**Figure 12 sensors-21-04925-f012:**
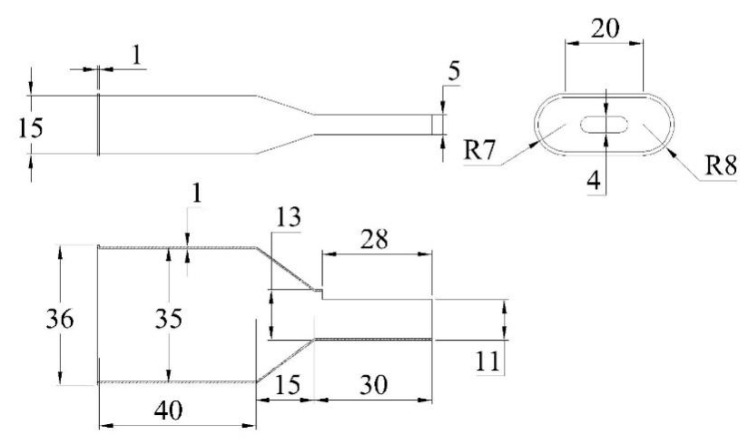
Orthographic views of the venturi (unit: mm).

**Figure 13 sensors-21-04925-f013:**
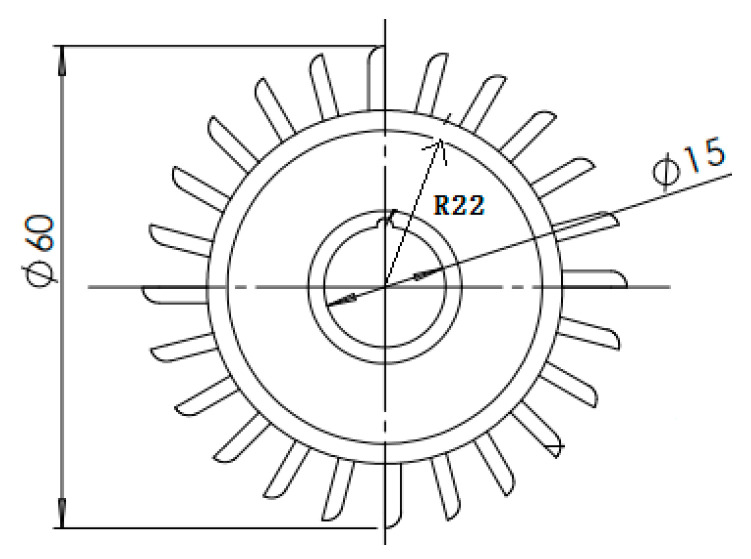
Dimensioned drawing of the water wheel (unit: mm).

**Figure 14 sensors-21-04925-f014:**
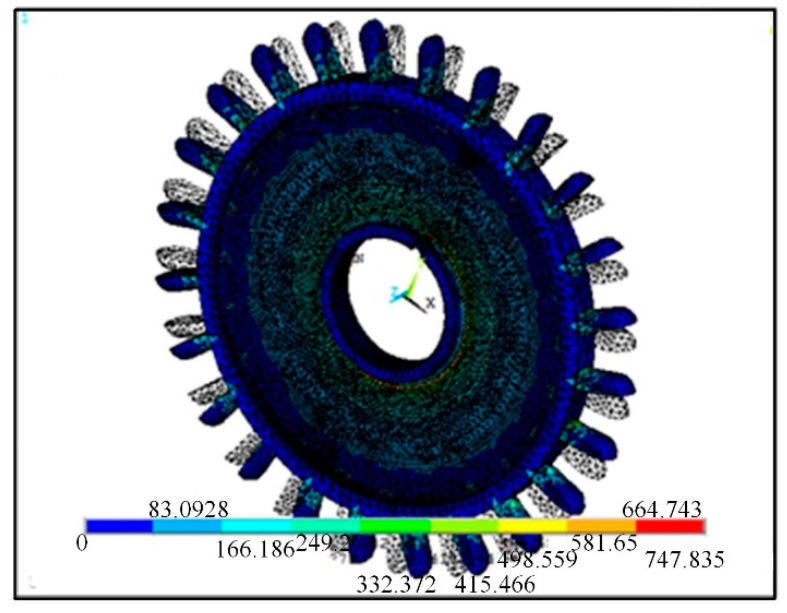
The strain of the water wheel.

**Figure 15 sensors-21-04925-f015:**
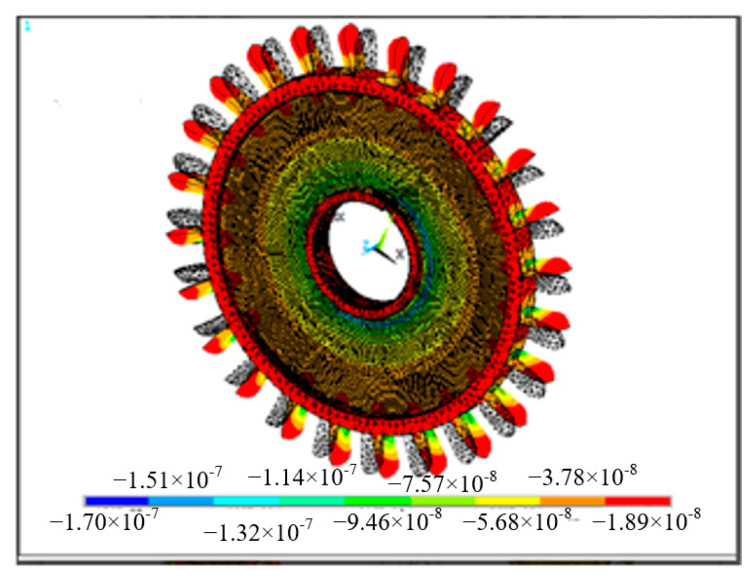
The stress of the water wheel (unit: MPa).

**Figure 16 sensors-21-04925-f016:**
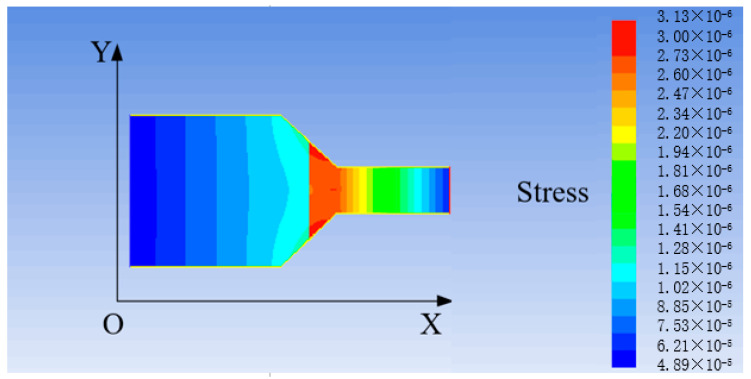
Stress of the venturi (Unit: Pa).

**Figure 17 sensors-21-04925-f017:**
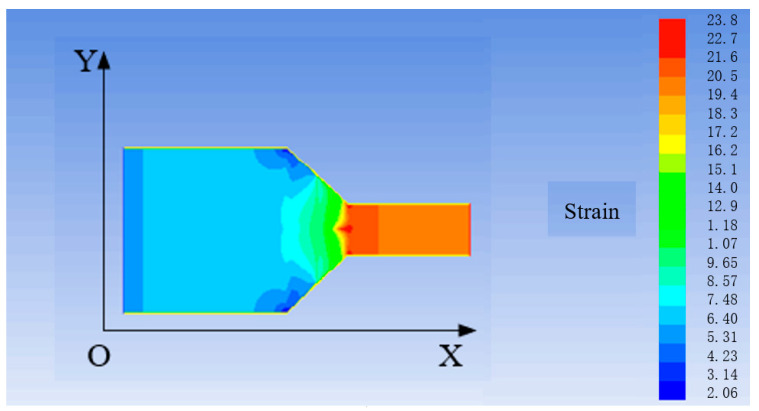
Strain of the venturi.

**Figure 18 sensors-21-04925-f018:**
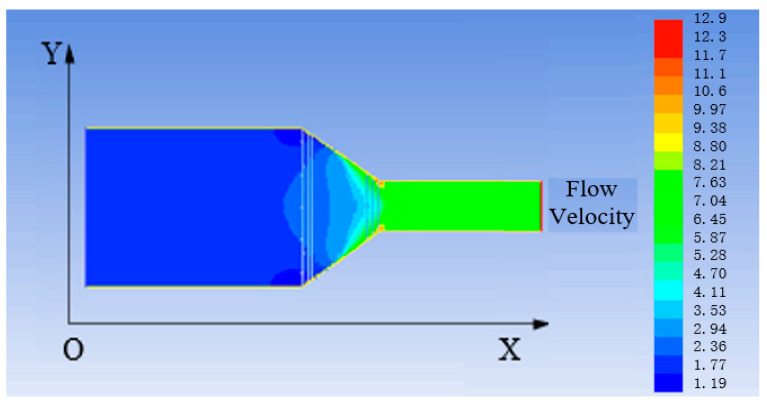
Change of the flow velocity (unit: m/s).

**Figure 19 sensors-21-04925-f019:**
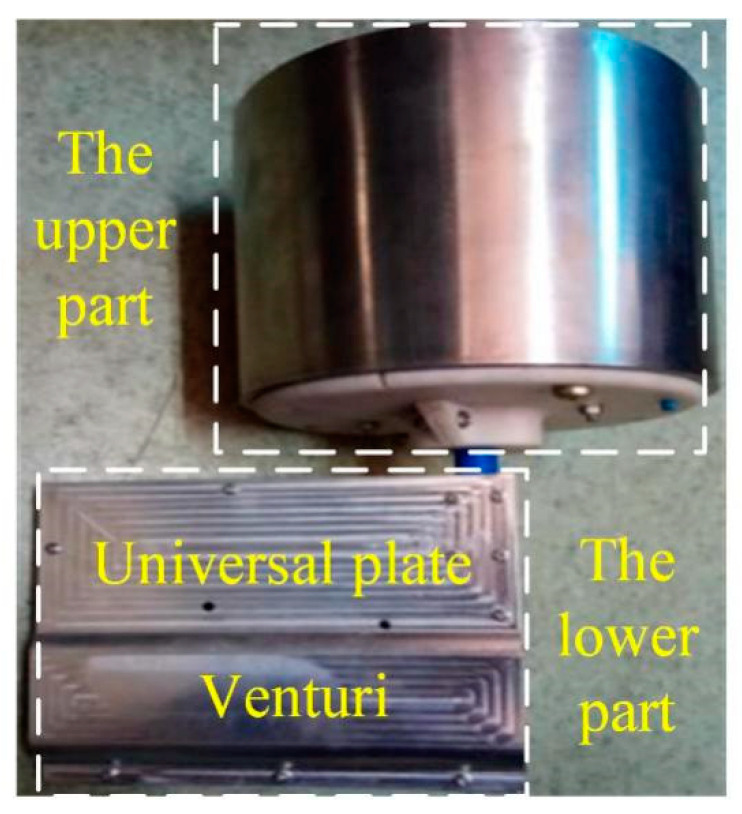
Sensor for monitoring flow direction and velocity.

**Figure 20 sensors-21-04925-f020:**
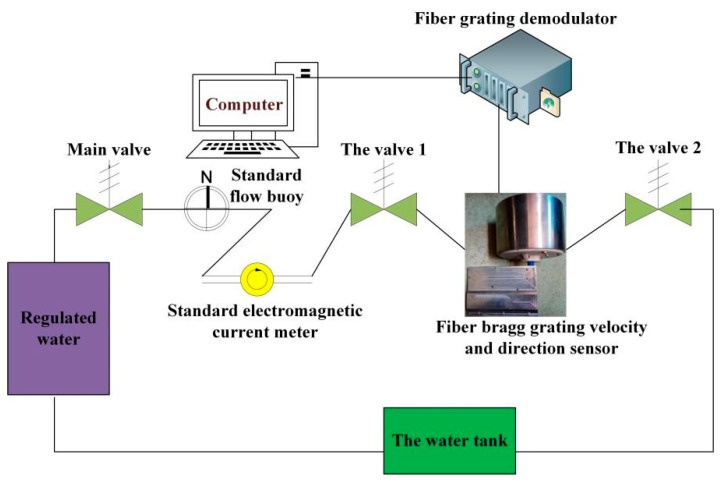
Experimental scheme of the sensor to monitor flow direction and velocity.

**Figure 21 sensors-21-04925-f021:**
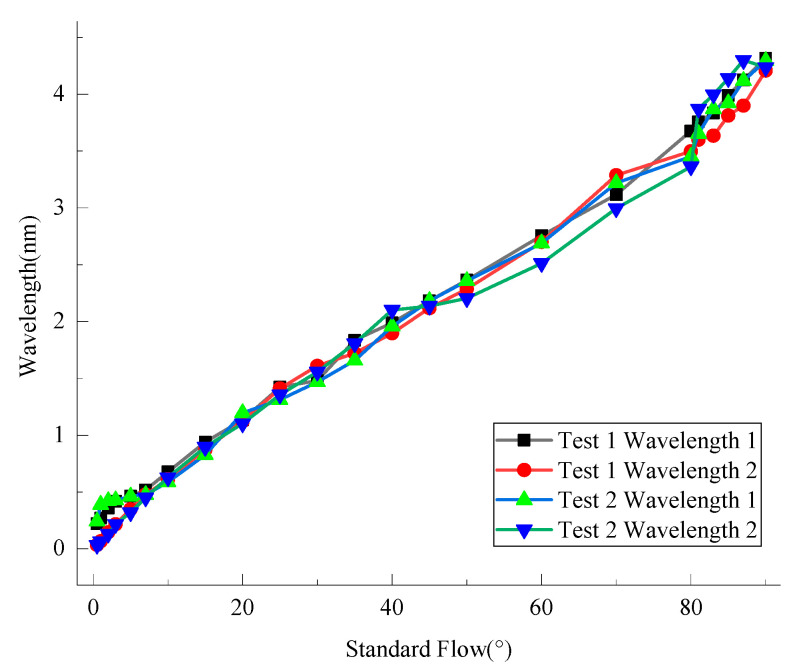
Wavelength increment and flow direction.

**Figure 22 sensors-21-04925-f022:**
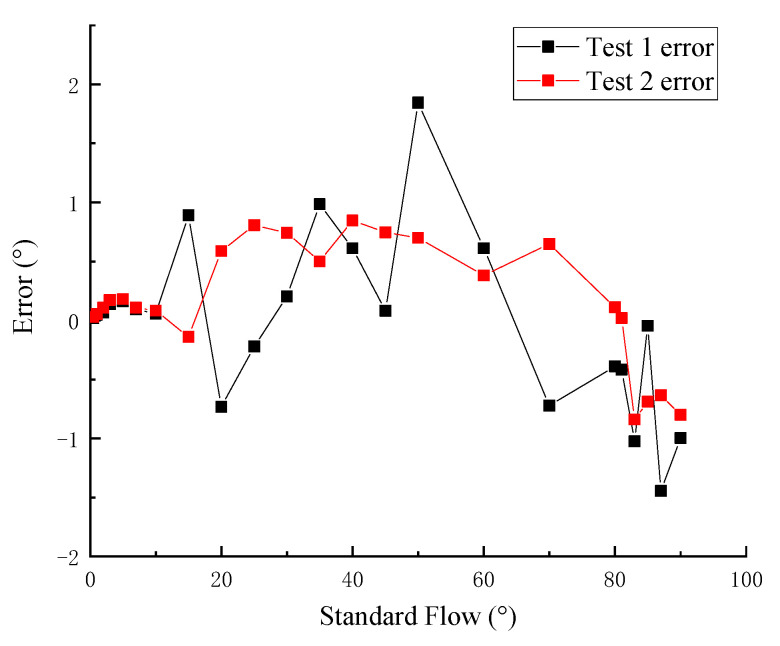
The error.

**Figure 23 sensors-21-04925-f023:**
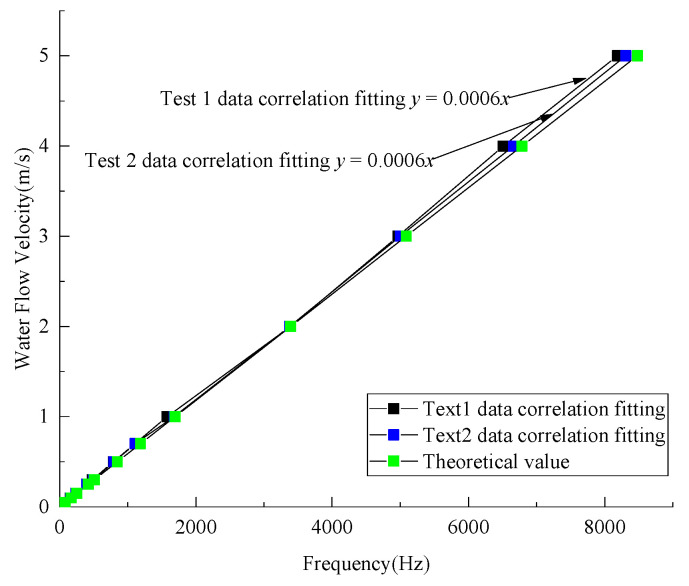
Analysis of the experimental frequency and the theoretical frequency.

**Figure 24 sensors-21-04925-f024:**
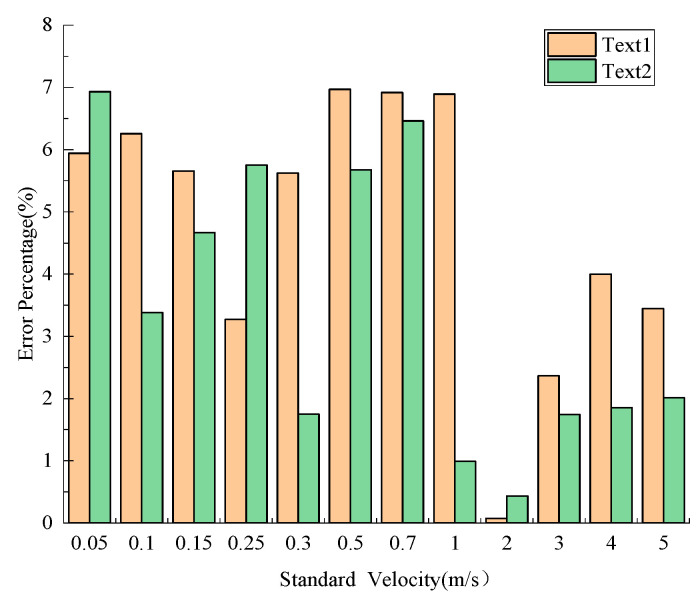
The comparison analysis of error range.

**Table 1 sensors-21-04925-t001:** The parameters of the FBGs.

Number of Grating	Purpose	Central Wavelength (nm)	3db Bandwidth (nm)	Reflectivity	The Length of Grating
1	Direction text	1533.00	0.26	93.834%	10 mm
2	Direction text	1550.96	0.27	91.946%	10 mm
3	Direction temperature compensation	1539.90	0.22	91.290%	10 mm
4	Velocity text	1544.93	0.26	93.347%	10 mm
5	Velocity temperature compensation	1554.97	0.22	90.450%	10 mm

## Data Availability

Data sharing is not applicable to this article.

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
