# Peer review of "Design of Flow Velocity and Direction Monitoring Sensor Based on Fiber Bragg Grating"

_sensors, 2021, doi:10.3390/s21144925_

Round 1
Reviewer 1 Report
This manuscript described a flow velocity and direction sensor based on fiber Bragg grating(FBG). The structure of the sensor is designed and simulated by finite element analysis. The experiment results are analyzed. Although the author gives a relatively complete manuscript, there are there are still two points that need to be specified, otherwise it is not suitable for publish in Sensors.
- First of all, the necessity of using fiber Bragg grating should be explained. FBG sensors have been widely used in many fields. This technology is not novel. Therefore, the authors should clarify the advantages or necessity of FBG sensors in this measurement.
- 5 FBGs are applied in the experiment. Some of them are used as temperature compensation sensors. However, the temperature information are not mentioned in this manuscript. How does the temperature compensation sensor work ? How are the measurements demodulated ? The authors should be explained in details.
Reviewer 2 Report
The authors present an intricate sensor design to mechanically couple an external fluid flow to several FBGs whose measured strain in turn encodes the fluid flow magnitude and direction. Components of this design are validated through mechanical analysis and FEM methods, which ultimately leads to experiments performed on a sensor prototype.
Whereas the design is presented clearly, I see a few areas of improvement that the authors may wish to consider. Perhaps they could clarify the following points in their manuscript.
Major comments
l57-59. Please rephrase/omit the novelty claim that this is the first sensor to measure both flow magnitude and direction using FBG sensing. Although uncommon, this is not the first time a FBG based sensor has been presented for measuring fluid flow magnitude and direction. See the references below [1, 2, 3].
Please remark on the potential effect of the combined spring formed by 'Spring 1' and 'Metal wire 1'. What effect, if any, should this have on the strain measured by 'FBG 1'? Note that this is a separate issue from the pasting technology and strength point raised in 4.3.1.2 and 4.3.1.3.
The choice for a relative error metric in Figure 22 is confusing. Why not list the absolute error in degrees? Surely the absolute uncertainty for flow direction is more relevant than the relative error. Additionally, this graph does not allow the reader to assert whether there's a systematic dependency of the direction error depending on the flow direction.
Regarding an observation from figure 23. Can you confirm that you have measured the top shaft to have vibrated at 8kHz, according to Eq 14? 
If so, I would expect serious mechanical consequences to the sensing unit. Surely in the range of 0 to 8kHz some mechanical resonance must have been observed and compensated for.
Additionally, it is unclear at what rate the FBGs are sampled by the interrogator unit. Modern interrogators achieve up to 5kHz which is not enough to reach the Nyquist frequency for your intended signal frequency.
Finally, a claim of a high accuracy (abstract, line 25) is at its worst 'wrong' and at its best 'overly positive' for an error of 6% and 7% for flow magnitude and direction respectively.
Minor comments
Please insert spaces between words and references.
l42. Please refer to Acoustic Doppler Current Profiler (ADCP) instead of the brand Aquadopp.
l183. Please define TGG
The righthand side diagram of Figure 11 seemingly does not have the lines O1' - R (top) and O2' - R (bottom) 90 degrees apart. Shouldn't that be the case?
l355-357. The signal processing not clearly described. What kind of filtering/window has been applied to transform the sampled periodic signal into a signal amplitude value as displayed in Figure 21?
Figure 23. Please change the legend key for 'Text1' and 'Text2'.
Additional References
[1] Jewart, C.; McMillen, B.; Cho, S.K.; Chen, K.P. X-probe flow sensor using self-powered active fiber Bragg gratings. Sensors and Actuators A: Physical 2006, 127, 63-68. doi: 10.1016/j.sna.2005.12.024
[2] Lu, P.; Chen, Q. Fiber Bragg grating sensor for simultaneous measurement of flow rate and direction. Meas. Sci. Technol. 2008, 19, 125301. doi:10.1088/0957-0233/19/12/125302
[3] Wolf, B.J.; Morton, J.A.S.; MacPherson, W.N.; van Netten, S.M. Bio-inspired all-optical artificial neuromast for 2D flow sensing.
Bioinspir. Biomim. 2018, 13, 026013. doi:10.1088/1748-3190/aaa786.
Round 2
Reviewer 1 Report
The new manuscript has been improved and can be published in Sensors.
This manuscript is a resubmission of an earlier submission. The following is a list of the peer review reports and author responses from that submission.